# Impact of the Built Environment and the Neighborhood in Promoting the Physical Activity and the Healthy Aging in Older People: An Umbrella Review

**DOI:** 10.3390/ijerph17176127

**Published:** 2020-08-23

**Authors:** Guglielmo Bonaccorsi, Federico Manzi, Marco Del Riccio, Nicoletta Setola, Eletta Naldi, Chiara Milani, Duccio Giorgetti, Claudia Dellisanti, Chiara Lorini

**Affiliations:** 1Department of Health Sciences, University of Florence, Viale GB Morgagni 48, 50134 Florence, Italy; guglielmo.bonaccorsi@unifi.it (G.B.); chiara.lorini@unifi.it (C.L.); 2Postgraduate School in Hygiene and Preventive Medicine, University of Florence, Viale GB Morgagni 48, 50134 Florence, Italy; marco.delriccio@unifi.it (M.D.R.); chiara.milani@unifi.it (C.M.); duccio.giorgetti@unifi.it (D.G.); 3Department of Architecture, University of Florence, Via della Mattonaia, 14, 50121 Florence, Italy; nicoletta.setola@unifi.it (N.S.); eletta.naldi@unifi.it (E.N.); 4Department of Epidemiology, Regional Health Agency of Tuscany, Via P Dazzi, 1, 50141 Florence, Italy; claudiadellisanti70@libero.it

**Keywords:** built environment, neighborhood, healthy aging, physical activity, elderly, walkability, health promotion, older adults

## Abstract

(1) Background: The aim of this study is to establish which specific elements of the built environment can contribute to improving the physical activity of self-sufficient, noninstitutionalized and living in the city adults > 65 years. (2) Methods: An extensive literature search was conducted in several database. Umbrella review methodology was used to include the reviews that presented a sufficient methodological quality. (3) Results: Eleven reviews were included. The elements positively associated with physical activity in older adults were: walkability; residential density/urbanization; street connectivity; land-use mix-destination diversity; overall access to facilities, destinations and services; pedestrian-friendly infrastructures; greenery and aesthetically pleasing scenery; high environmental quality; street lighting; crime-related safety; traffic-related safety. The elements that were negatively associated with physical activity were: poor pedestrian access to shopping centers; poor pedestrian-friendly infrastructure and footpath quality; barriers to walking/cycling; lack of aesthetically pleasing scenery; crime-related unsafety; unattended dogs; inadequate street lighting and upkeep; traffic; littering, vandalism, decay; pollution; noise. (4) Conclusions: Evidence shows that specific elements of the built environment can contribute to promoting older people’s physical activity. The city restructuring plans should take into consideration these factors.

## 1. Introduction

As stated by the Copenhagen Consensus Conference Statement 2019, “being physically active is a key factor in maintaining health and in normal functioning of physiological systems across the life-course” [1]; healthy cities play a fundamental role in promoting the healthy aging of older people, as declared by the 9th Global Conference on Health Promotion [2]. The WHO defines Healthy Ageing “as the process of developing and maintaining the functional ability that enables wellbeing in older age”. Functional ability is about having the capabilities that enable all people to be and do what they have reason to value, and it is made up of the intrinsic capacity of the individual, of the relevant environmental characteristics and the interaction between them [3].

The environment is made up of the “physical and social characteristics in which people live” and it represents one of the factors that most influences the life of each person [4]. On the one hand, the environment can have a direct effect on people’s health, through various types of pathological determinants (for example, air and water pollution, noise, electromagnetic pollution, infections, carcinogens, etc.). On the other hand, it can also affect people’s health in a more subtle and indirect way, affecting the behaviors and activities of the people who live there: for example, urban sprawl and the development of the periphery, associated with the spread of the car, have determined the development of sedentary behaviors, thus promoting the increase of diseases related to physical inactivity, such as obesity, diabetes, cardiovascular diseases and tumors [5]. Population ageing is about to become one of the most significant social transformations of the twenty-first century, with implications for many sectors of society. According to the 2019 Revision of World Population Prospects, by 2050, one in six people in the world (16%) will be over age 65, up from one in 11 (9%) in 2019, and the number of persons aged 80 or over is projected to triple. In addition, one in four people living in Europe and North America could be aged 65 or over [6]. This longevity revolution is interconnected with another massive trend concerning urban population growth: the world’s population living in urban areas is expected to increase from 55% in 2018 to 68% by 2050, which means that the world’s urban population will nearly double. In 2018 the most urbanized regions were North America (82%), Latin America (68%) and Europe (74%), where the level of urbanization is expected to increase to almost 84% in 2050 [7]. These demographical, epidemiological and geographical transitions move our attention to older people considered in the local urban context of the neighborhood. For these evident relationships between environment and health, since the beginning of the new century, collaborations between urban planning/architecture and public health have increased, with the aim of finding solutions to problems such as physical inactivity, obesity and mental illness (in terms of social exclusion, lack of social participation and sense of cohesion) [8].

The built environment is defined as the physical space of the environment which is human-made or modifiable and where people live and carry out their daily activities. It includes buildings (homes, schools, workplaces), open spaces (parks, recreation areas) and infrastructures (transportation systems). Within the built environment, the neighborhood can be identified as the geographical area immediately around the residence of a family, delimited by physical characteristics of the environment such as roads, rivers, train tracks. Neighborhoods generally have a strong social component (social interactions between neighbors, a sense of shared identity) and specific socioeconomic characteristics [9]. In old age, the neighborhood assumes particular importance because many older people spend much of their time there. Besides, there is a decrease in the range of action, especially for older adults, and the opportunities for interaction and meeting in the neighborhood therefore become increasingly important with advancing age [10]. This type of attachment to place, a relevant concept of environmental psychology, seems to be important in creating a sense of belonging for older adults, which determines physical and mental well-being [11,12,13].

The built environment has become the subject of increasing attention in recent years regarding its role in encouraging physical activity, as walking around the neighborhood is the most common type of physical activity for older people, and every day it is possible to reach the recommended levels of physical activity by practicing simple activities such as walking and cycling. It would be crucial, though, to design a community that could support the possibility to walk or cycle and to provide access to recreational services. Several studies have found that the characteristics of the neighborhood are strongly related to the increase in walking and physical activity rates [14,15,16,17]. Previous evidence shows that a built environment that supports and encourages physical activity has long-term effects on a large portion of the population and also facilitates the maintenance of good acquired habits [18]. Finally, the WHO “Global Recommendations on physical activity for health” [19] state that all healthy adults aged 65 years and above, unless they have specific medical conditions, benefit from physical activity in terms of:lower rates of coronary heart disease, hypertension, stroke, diabetes, colon and breast cancera higher level of cardiorespiratory and muscular fitnesshealthier body mass and composition and enhanced bone healthhigher levels of functional health, a lower risk of falling, and better cognitive function.

In adults aged 65 and over, physical activity can include leisure time physical activity or active travel for transportation, including both activities of mainly walking or cycling [20,21]. In particular, older adults should do at least 150 min of moderate-intensity or 75 min of vigorous-intensity aerobic physical activity throughout the week.

The aim of the present umbrella review, which includes systematic, literature, narrative and scoping reviews about the built environment and physical activity in older adults, is to describe the features of the neighborhood that can influence physical activity levels of self-sufficient, noninstitutionalized and living in the city adults aged more than 65 years.

## 2. Materials and Methods

The present study has been conducted following Aromataris et al. in the Joanna Briggs Institute Manual to conduct an umbrella review [22]. The methodology used is different from that of Cochrane for an overview of reviews, because we included not only the Cochrane intervention reviews produced by individual Cochrane review groups, but also other reviews that met the inclusion criteria, and is, therefore, more inclusive [23].

### 2.1. Inclusion Criteria

PICOT (Population, Intervention, Comparison, Outcome, Type of Study) [24] scheme was used to identify the papers that met the inclusion criteria: (1) peer-reviewed studies; (2) studies on older adults (it is important to note that there are different thresholds to define older adults: eligible reviews have to include a population older than 55 years old); (3) studies assessing the associations between built environmental factors and physical activity and (4) studies assessing the effectiveness of intervention on built environment in improving physical activity and movement; (5) type of studies: reviews (Table 1).

Reviews that included studies considering an independent, autonomous, nonhospitalized older population were considered eligible.

### 2.2. Search Strategy

PubMed/MEDLINE, EMBASE, Cochrane Library, Scopus, Avery Index, Sage Journals, Web of Science, and Health Evidence databases were searched (search strings in the appendix—Appendix A) up to 31 July 2019 for reviews investigating the specific interventions on urban elements of the built environment that could promote the physical activity in the older adults. The reference lists of the relevant reviews were also manually-searched for additional articles missed by the electronic search. The research was completed with the consultation of the general research engines (Google, Google Scholar). No data range in the search process was considered.

### 2.3. Study Selection and Data Extraction

The search query identified 2786 articles (939 in PubMed, 305 in EMBASE, 745 in Scopus, 17 in Chart Avery Index, 2 in Cochrane Library, 62 in Health Evidence, 428 in Sage journals, 288 in Web of Science). Two independent investigators (CD, EN) screened the titles, the abstracts of the identified records and the full text of the potentially eligible articles. In case of discrepancy, a third investigator (CL) was consulted until agreement was reached. After screening all the articles, 33 articles were selected, and 11 after removing duplicates. In the end, 11 reviews met the inclusion criteria.

Figure 1 represents the PRISMA (Preferred Reporting Items for Systematic Reviews and Meta-Analyses) flow-chart process of study selection [25]. We developed a summary table to record the characteristics of the included studies and the key information relevant to the research question. We extracted, summarized, and tabulated the following key information from each publication: author and year of publication, type of study and population, number and type of included studies, the built environment features considered and the tools used to measure them, the outcomes studied and the relative tool of measure, the method used to establish the relationship between the neighborhood features and the outcomes (if specified), and the quality evaluation of the reviews (Table 2).

The methodological quality of the selected reviews was assessed by using the Health Evidence “Quality Assessment Tool—Review Articles”. The Quality Assessment Tool—Review Articles is a 10-item instrument related to essential features of the methodological rigor across reviews, that produces a final score ranging from 0 to 10: higher scores indicate higher quality. The scores can also be ordered as strong (8–10), moderate (5–7) or weak (4 or less) [26].

### 2.4. Data Synthesis and Analysis

In order to obtain a summary of the results from the included reviews, we first extrapolated the types of physical activity that were considered by the works; in this way, different types of physical activity have been identified:Overall physical activity (PA), when the activity was not otherwise specifiedLeisure time physical activity (LTPA)WalkingActive travel (AT), which means a mode of transport which involves physical activity to get from one destination to another [27].

Then we attributed to each type of physical activity the associations with the elements of the built environment found by the reviews included.

## 3. Results

### 3.1. Description of the Included Studies

After the screening, 11 reviews were included. Many of the included works are systematic reviews, with or without meta-analysis, except from Cunningham and Michael (2004) [28], and Tuckett et al. (2018) [29], whose works are literature reviews, and Levasseur et al. (2015) [30], who produced a scoping review. The included articles are of good methodological quality; for the qualitative evaluation, nine were categorized as “strong quality”, two were categorized as “moderate quality” (Table 2).

The total number of primary studies included in the 11 reviews was 682.

Almost all of the studies met all of the PICOT criteria, even if there were some exceptions. Among the exceptions, only 9% of the included studies referred to older adults in the work of Gadais et al. (2018) [31] and only six studies out of 27 in Cunningham and Michael (2004) [28]. Won et al. (2016) [32] considered only neighborhood safety as influencing both health behaviors, such as physical activity and walking, and health outcomes, such as health status, mental health, physical function, and morbidity/mortality, which were excluded from our analysis that considered 16 out of 32 articles; Levasseur et al. (2015) [30] also considered social participation and only 39 out of 50 were focused on mobility; Yen et al. (2009) [33] considered various health outcomes, including mental health, health behaviors, morbidity and mortality, so only 7 out of 33 included studies analyzed physical activity as the outcome.

Table 2 presents the summary of the features considered of the included reviews.

Concerning the type of studies included in the reviews, they are observational studies (the majority of them are cross-sectional, with a dearth of longitudinal ones), quasi-experimental studies, and mixed studies, using both quantitative and qualitative methods.

### 3.2. Built Environment Features

Some of the reviews report the way the studies they included define the neighborhood or area of investigation. The neighborhood is defined in heterogeneous ways. When it is objectively described, it refers to an administrative or census area (e.g., postal code), or by buffer radii ranging from 100 m to more than 1 km (mostly <1 km); when it is defined by the study participants, it depends on their perception: walking minutes from home (mostly 10–20 min), individual characteristics of interest, perceived boundaries. Many studies in the included reviews did not even report how they defined the neighborhood (Appendix B).

As said in the methods, the features of the built environment (“interventions” in the PICOT model), include many aspects. Some authors [20,21,34] classify physical environmental variables according to Neighborhood Environmental Walkability Scale (NEWS), which include the category “walkability” as the main and most generic attribute, and other six categories in its expanded form: residential density/urbanization, street connectivity, access to/availability of services/destinations, pedestrian/cycling infrastructure and streetscape, aesthetics and cleanliness/order, safety and traffic. Some of the categories include subcategories, as partly shown in Table 2.

The other authors use different categories to classify the elements of the built environment, splitting out the features of the physical environment in Land-use patterns, Urban design characteristics and Transportation systems [31], not categorizing the elements [29] or considering only safety domains of the neighborhood [32]. Levasseur et al. (2015) [30] analyzed, organized and synthesized data according to the International Classification of Functioning, Disability and Health (ICF), which classifies the features of the environment into five domains: Products and technology; Natural environment and human-made changes to environment; Support and relationship, Attitudes and Services, systems and policies. Moran et al. (2014) [35] categorize so as to merge the related environmental factors that emerged into subthemes and themes, named using content-characteristic words. The five themes are: Pedestrian infrastructures, Safety, Access to facilities, Aesthetics, and Environmental conditions. Yen et al. (2009) [33] describe each study in terms of six possible types of neighborhood exposure measures: socioeconomic composition, racial composition, demographics, social environment, perceived resources and/or problems and physical environment. We took into consideration only the last two measurements. Cunningham and Michael (2004) [28] start from the key feature of the built environment according to planning literature: Transportation system, Land-use pattern, Density, Land-use mix, Street connectivity, Aesthetic quality, Connectivity and Microscale elements.

Environmental features were assessed using objective or perceived measurement, and sometimes a combination of the two. Some reviews mentioned the instruments used to measure the built environment [32,33,34,35]. In particular, Barnett et al. (2017) [34] and Cerin et al. (2017) [20] provided a detailed list of which measuring instrument had been used in each included article, which helped in defining an overview of the most used methods and tools (Appendix B). NEWS questionnaire appeared to be the most used as the perceived way to assess the built environment, and whether the use of Geographic (GIS) also accompanied the other systematic observational methods or datasets represents a diffuse objective measurement.

### 3.3. Outcome Measures

As said in the methods section, the outcome of this paper was the promotion of physical activity (PA) in older people, which included, according to the reviews, different types of activities. Van Cauwenberg et al. (2018) [21] analyzed the promotion of physical activity during leisure-time (LTPA): leisure-time walking, leisure-time walking within the neighborhood, leisure-time cycling, leisure-time walking and cycling combined, and overall LTPA. Barnett et al. (2017) [34] considered, as outcomes, only total physical activity (PA) and walking, which were considered as distinguished typologies of activities. Cerin et al. (2017) [20] focused on active travel (AT): total walking for transport, within-neighborhood walking for transport, combined walking and cycling for transport, cycling for transport, and all AT outcomes combined. Gadais et al. (2018) [31] analyzed, as outcomes, physical activity and active travel. Won et al. (2016) [32] considered physical activity and walking. Levasseur et al. (2015) [30] considered, as outcomes, mobility, that has been relocated within the general category Physical activity (PA). Other authors [28,29,33,35] primarily considered PA. Finally, Van Cauwenberg et al. (2011) [36] analyzed PA promotion, distinguishing recreational physical activity, walking, cycling. Recreational activity has been relocated in the category Leisure Time Physical Activity (LTPA), while walking has been considered as an independent category.

### 3.4. Quality Assessment and Moderator of Associations

Some reviews have also described the method used to establish the strength of the association between environmental factors and physical activity, summarizing the evidence emerging from the various included studies (Table 3).

Van Cauwenberg et al. (2018) [21], Barnett et al. (2017) [34], and Cerin et al. (2017) [20] used the meta-analytical approach explained in Cerin et al. (2017) [20], which statistically quantifies the strength of the tests for “Environment–PA” associations. In the synthesis process the *p* values are estimated for each combination of environmental attribute and outcome (PA), taking into account the sample size and the quality scores of the articles.

Levasseur et al. (2015) [30], Moran et al. (2014) [35] and Van Cauwenberg et al. (2011) [36] simply summed the associations between PA and elements of the built environment of the individual works.

The other works do not report how they established the strength of the associations, even if Won et al. (2016) [32] assessed the studies for a methodological quality using the assessment tool adapted from the Effective Public Health Practice Project (EPHPP), and Yen et al. (2009) [33] assessed the studies using a set of criteria created specifically on the basis of previous comments on research on the health of the neighborhood.

Finally, some reviews mention the importance in considering confounding factors which might be responsible for some of the results, with the neighborhood-level Socioeconomic Status representing the strongest and most consistent predictor of variety in outcomes. In particular, three reviews [20,21,35] outlined findings from articles that have examined moderating effects on environmental correlates of physical activity. The moderators of associations can modify the degree of the association between the elements of the built environment and the physical activity in the people aged >65 years, and the direction varies depending on the considered moderator. Several but inconsistent individual—and environmental—level moderators of associations were identified:individual moderators: sociodemographic (age, sex, level of education, income, employment status, race/ethnicity, marital status), health status/functionality, psychosocial factors, duration of residency, vehicle ownership or driving status;environmental moderators: area-level socioeconomic status, residential density/urbanization, pedestrian/cycling infrastructure and streetscape, aesthetics and cleanliness/order, safety and traffic, geographical scale, neighborhood definition).

According to the analysis, we focused our attention on considering, as environmental moderator, the influence of some specific environmental definition: the country or continent of investigation, the setting intended as urban or local area and the neighborhood definition. The majority of studies included in the reviews are located in developed countries, especially in North America, which represent the only geographical area of interest for many reviews [28,31,32] and where the majority of studies, between 40% to 86%, are placed in the other reviews. The second most diffuse continent is Europe, generally representing around 20% of the studies and reaching its maximum at 35% in Moran et al. [35]. Oceania was in third place, followed by Asia. The other component strictly connected to our research is represented by the urban or suburban context. The studies are placed mostly in urban areas, or in mixed contexts, but there is no shortage of rural setting studies. It is important to note that some significant relationships emerged only for people living in rural or urban areas.

### 3.5. Findings: Associations between Outcomes and Built Environment Features

The included reviews reporting many associations between physical activity in older people and elements of the neighborhood (Table 2). In analyzing the results, a sum of the findings has been made. As stated in the methods, in summarizing the associations between elements of the built environment and physical activity in older adults, all the elements analyzed by the reviews included were taken into account; then each of the four physical activity typologies were associated with the features that influence it positively or negatively (Table 3).

## 4. Discussion

The attempt to implement an umbrella review was prompted by the need to summarize the importance of certain elements of the built environment in promoting physical activity in older adults. The aim was to provide evidence for policymakers on what elements are strongly associated with an improvement of leisure time, walking or exercise, in order to carry out related policies. To fulfill this aim and to make better use of and improve data, we decided to resume the available evidence and then to highlight the specific features of the built environment that have found favor in physical activity levels in older people.

Physical activity is used as an umbrella term that includes both structured and unstructured forms of leisure, transport, domestic and work-related activities, and it entails body movement that increases energy expenditure relative to rest [1].

Aging is associated with a decreased efficiency of different cognitive functions as well as in perceptive, physical and physiological changes. Physical activity can positively affect the physical [37] and cognitive efficiency and mental health of older healthy individuals, and possibly reduces the risk of progression into dementia [38] and depression [39].

### 4.1. Built Environment Evidences

According to the results of this umbrella review, some elements of the built environment emerge as quite clearly positively associated with the promotion of PA, including overall access to facilities, destinations and services such as public transport, recreational facilities, the presence of user-friendly infrastructure and the most general practicability of the neighborhood [30,32,33,35].

The low presence and poor quality of pedestrian access to shopping centers, footpaths and sidewalks and the presence of traffic, pollution, noise and crime-related events were negatively associated with the increase in overall physical activity [28,29,30,33].

The same elements were also positively associated with the other categories of physical activity:

Walking, as well as LTPA, is positively affected by general walkability and by the possibility of accessing open spaces, shops and services and facilities [21,34].

Active Travel is positively associated not only with the presence of services and structures but also with the presence of specific structures that facilitate movement, such as pedestrian paths, cycle paths, and pedestrian and cycle structures [20].

While the Leisure Time Physical Activity is negatively associated with the presence of barriers for walking and cycling, in Active Travel there is a negative association with the decay of the built environment (decay and vandalism).

The evidence that emerges from the analysis of the results of this umbrella review is not clearly conclusive, since the works included are extremely uneven in the definition of the interventions and outcomes and the tools to measure them, as shown in the Table 2. This lack of homogeneity makes it difficult to compare results and obtain strong and univocal evidences. Moreover, for reasons of public spending and setting, it is practically impossible to carry out experimental studies focused on the impact of the built environment on the promotion of the physical activity at the neighborhood scale.

Despite this lack of homogeneity and despite the lack of experimental studies, the indications that come from the included papers can certainly suggest some intervention priorities in terms of urban planning.

In fact, the relationship between built environment and PA for older people is widely discussed in scientific literature (11 reviews have been included in this umbrella, for a total of 682 primary studies) and the results that derive from these studies allow us to give important evidence to support politicians, administrators and policy makers: the general walkability of the neighborhoods, the presence of safe paths for pedestrians and cyclists (sidewalks, cycle paths, pedestrian areas) and the access to structures such as shops and commercial, infrastructures such as public transport and spaces such as parks and recreational places, all have an important impact in promoting physical activity in older people.

Future studies should focus on the homogenization and systematization of the measurement tools of the outcomes and interventions, in addition to giving them a more univocal definition; this should allow for giving stronger evidence of the role of the specific features of the built environment in promoting both PA of older people and healthy aging.

### 4.2. The Importance of the Local Context

The results and considerations that emerged from the reviews highlight the importance of considering a wide range of degrees in analyzing the scale of the built environment, ranging from general issues to specific elements, such as the presence of benches or the footpath quality. The close environment, as a setting of everyday life, acquires particular relevance for this category of users, therefore their way of living is profoundly rooted in the micro and mesoscale physical and psychological component. In fact, due to the peculiarities of the lives of older people, mostly hinged on slow mobility and not necessarily rotating around work activities, the influence of proximity should not be underestimated. The intent of the study to extrapolate global recommendations, generally valid for this category of space users, must take into account the importance of some moderators of association connected to the physical environment, which are correlated to urban components.

The urban context represented our sight of interest due to the epidemiological and geographical transition connected to chronic disease, and its local dimension is primarily influenced by the country or continent in which it is placed. Indeed, the importance of considering some peculiarities as mostly connected to the specific geographical, cultural and political area should be considered.

Some results are related to specific countries and continents revealing why some reviews focused only in specific geographical areas, recognizing some peculiarities, such as discrepancies when considering safety issues or the prevalence in the US of car-dependent landscapes or more numerous crimes and traffic accidents than in most other high-income nations. The same way of thinking in limiting the generalizability of findings should be adopted when considering Western cities compared to the built environments of Africa, Asia and South America, so caution should be paid when translating country-focused findings to other countries.

Moreover, it is important to consider the way of measuring perspective and needs related to physical activity and built environment. In general, the associations tend to differ concerning the kind of measurement done (objective or perceived) both to environmental issues and physical activity. Some studies showed stronger associations when using objective measures and vice-versa. This suggests that objective and perceived measures may be differentially related to different factors, and it would be useful to include perceived as well as objective measures in future studies. Built environmental attributes relate differently to different behaviors (for example attributes within safety and aesthetics domains are more subjective in their interpretation and thus depend on individuals’ perceptions, while aspects related to destinations and services are more objective and so less susceptible to interpersonal differences in perceptions). The difference may also relate to common method bias associated with self-reported environmental features and physical activity and also to not tailored use of objective measurement (e.g., the wrong accelerometer cut-point). Self-reported measures are more likely to be influenced by culture and, thus, yield different findings across geographical locations due to measurement rather than substantive reasons and take into account ones’ attributes. It is important to evaluate the more appropriate method of measurement in relation to the factor and outcome addressed, for example, in recent years several tools have emerged that would help to reduce measurement error and clarify the impact on the local context for the objective assessment of environmental issues (e.g., Geographic Positioning System). In order to reveal environmental issues related to microscale architecture, the use of qualitative methods appear to be important: interviews and focus groups or spatial qualitative methods (observations, photo-voice, virtual reality experiment) are able to add depth and detail to the results. Specific details unique to older adults such as design quality of a bench in terms of comfort (sheltered in winter, shadowed in summertime) or usability (easiness to sit on and get up) reveal in-depth information not merely related to the general presence of a rest area, on how, what and why some environmental features are influence older adults’ physical activity. Therefore, the use of objective measurements in combination with self-reported data provides a more accurate understanding of environmental influences on physical activity. Additionally, the combination of interviews with spatial methods, by connecting specific objective environmental attributes to subjective experience, also provides more accuracy.

Finally, future studies will address the impact of local issues as moderators of the effects of the built environment in promoting physical activity. Indeed, the 11 studies included in this paper take into account the universal and general characteristics of the built environment, that is, those that can be found everywhere, but barely consider the effects of local phenomena such as climate. Interesting issues that should be investigated include whether the association between the ease of access to green areas and physical activity change, while considering extremely hot climates or extremely cold areas [40] and what the impact is of the natural environment in moderating the effect of the built environment interventions on promoting physical activity.

Besides that, the cultural influences and the sociodemographic characteristics of the local context could also influence these associations [41].

What emerges is a clear need to design studies that are able to explore these aspects at population and local levels using instruments and tools that could be systematically compared.

### 4.3. Limits of the Reviews Included in the Umbrella

We analyzed the reviews that included studies that differed in their study design, targeted population (for characteristics and sample size), setting of implementation schedule, duration, assessment and evaluation of interventions and outcomes. These differences in the primary studies included in the reviews represented serious obstacles in realizing the aim of this umbrella review.

Moreover, there is a lack of a clear definition of key elements such as “built environment”, “physical activity” and “walkability”: this makes it sometimes impossible to generalize the “real impact” of each experience or compare the contribution obtained by each of the studies.

### 4.4. Limits of Our Review

An umbrella review itself has limitations in its methodological process, such as the potential loss of information because of an excess of the synthesis of already-produced reviews.

Another possible limit is represented by the quality of the included primary studies on which reviews are built, as stated by the authors, as well as the strength of the conclusions of reviews themselves, and our umbrella review, too.

The large number of both the elements of the built environment and the outcomes, and consequently measurement instruments for them, implies a heterogeneity that makes it difficult to synthesize and compare the different conclusions.

## 5. Conclusions

This is the first umbrella review that makes a synthesis of the reviews produced on the effectiveness of interventions on the built environment to promote physical activity in older adults. Despite the mentioned limits, we can conclude that some aspects of the built environment are favored positively or negatively, in various forms, that affect the physical activity of older people.

Future research should find and use homogeneous tools and working methods to compare the different experiences so as to produce conclusive evidence. To make this, a first suggestion is to tailor studies on the older population in relation to specific urban elements, identifying priority areas, converging on specific elements to be analyzed, using the same measurement tools.

Another priority is the definition of a common language in relation to urban elements, outcomes, and measuring instruments. Some reviews included in the paper have already used this approach in order to synthesize planning elements and tools, so as to bring out associations with outcomes of the included studies [20,21,34].

Finally, as a medium-term proposal, we suggest, wherever possible, to modify the environment according to the strongest associations between physical activity and some of the factors that emerged from our review (e.g., Walkability, Overall access to facilities, Access to public transport, Access to nature/parks/open space, Pedestrian-friendly infrastructure) in order to improve healthy ageing.

## Figures and Tables

**Figure 1 ijerph-17-06127-f001:**
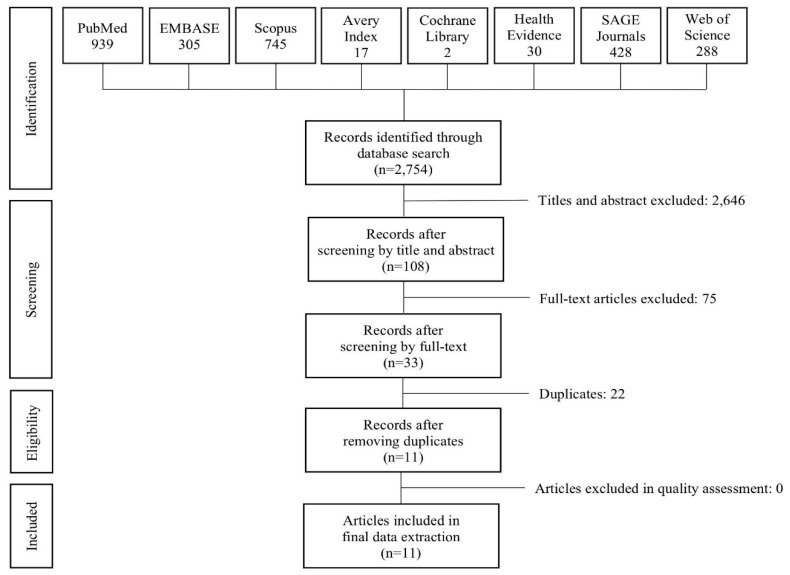
Flowchart of selection process of the reviews.

**Table 1 ijerph-17-06127-t001:** PICOT (Population, intervention, comparison, outcome, type of study) scheme to define inclusion criteria.

Parameter	Description
Population	Inclusion—Older adults, noninstitutionalized self-sufficient citizens Exclusion—Children, adolescents, young adults, institutionalized/hospitalized, housebound
Intervention	Features of built environment: street connectivity and grid pattern, road signs, easy access to the structures, parking for bicycles, street lights, presence of underpasses, walking trails, sidewalks, biking trails, structure and street security, reduction of road traffic, ad hoc structures (gyms, dancing halls), open spaces, benches, shaded areas, access to transport (bicycles), presence of handrails, presence of stairs and steps
Comparison	No intervention—Absence of the selected features of built environment
Outcome	Inclusion—Promotion of physical activity and movement Exclusion—Other health outcomes
Study design	Inclusion: review

Self-sufficient, independent older population was chosen because of the huge burden of disease and disability in elderly health care for health systems. Papers that specifically considered effects of built environment on physical activity in other subgroups were excluded. Qualitative and quantitative reviews that investigated whether specific interventions on urban elements of the public built environment could promote the physical activity in older adults were eligible for inclusion. As “intervention”, many features of the built environment have been considered: street connectivity and grid pattern, road signs, easy access to the structures, parking for bicycles, street lights, presence of underpasses, walking trails, sidewalks, biking trails, structure and street security, reduction of road traffic, ad hoc structures (gyms, dancing halls), open spaces, benches, shaded areas, access to transport (bicycles), presence of handrails, presence of stairs and steps. No outcome other than physical activity and movement was included.

**Table 2 ijerph-17-06127-t002:** Main elements of the included reviews.

Author(s) (Year)	Type of Study Population	Number and Type of Included Studies	Environmental Features/Factors	Environmental Features Measurement	Outcome (Physical Activity)	Physical Activity Measurement	Measure of Association	Conclusion	Qualitative Evaluation
Barnett et al. (2017)	Systematic review and meta-analysis Older adults ≥65 years old	100 Cross-sectional, longitudinal and quasi-experimental	NEWS categories: — Walkability — Residential density/urbanization — Street connectivity — Access to/availability of destinations and services — Infrastructure and streetscape — Safety	Total environmental attributes: Objective: 48%, Perceived: 52% Specifically: — Walkability: Obj. 11%, Perc. 2%; — Residential density/urbanization: Obj. 21%, Perc. 15%; — Street connectivity: Obj. 10%, Perc. 16%; — Access to/availability of destinations and services: Obj. 29%, Perc. 45%; — Infrastructure and streetscape: Obj. 12%, Perc. 34%; — Safety: Obj. 7%, Perc.: 40%.	Total physical activity (PA) and walking	Total PA outcomes: Objective: 27%, Self-reported: 74% Specifically: — Total PA: Obj. 8%, Self-rep. 23% — Total walking: Obj. 9%, Self-rep. 47% — Total MVPA: Obj. 15%, Self-rep. 14%	Meta-analytic approach + quality assessment	Positive association with total physical activity for: — Walkability (*p* < 0.001) — Overall access to destinations and services (*p* < 0.001) — Access to recreational facilities (*p* < 0.001) — Access to shops/commercial (*p* = 0.006) — Access to public transport (*p* = 0.016) — Access to parks/public open space (*p* = 0.002) — Walk-friendly infrastructure (*p* = 0.009) — Greenery/aesthetically pleasing scenery (*p* = 0.004) — Crime/personal safety (*p* < 0.001) Positive association with total walking for: — Walkability (*p* = 0.001) — Residential density/urbanization (*p* = 0.036) — Overall access to destinations and services (*p* = 0.009) — Access to shops/commercial (*p* = 0.001) — Access to public transport (*p* = 0.011) — Access to parks/public open space (*p* = 0.014) — Walk-friendly infrastructure (*p* = 0.042) — Street lighting (*p* = 0.042) — Greenery/aesthetically pleasing scenery (*p* = 0.002) — Crime/personal safety (*p* = 0.014)	10
Cerin et al. (2017)	Systematic review and meta-analysis Older adults ≥65 years old	42 Cross-sectional and longitudinal and quasi-experimental	NEWS categories: — Walkability — Residential density/urbanization — Street connectivity — Access to/availability of services/destinations — Streetscape and pedestrian/cycling infrastructure — Aesthetics and cleanliness/order — Safety and traffic	Total environmental attributes: Objective: 33%, Perceived: 68% Specifically: — Walkability: Obj. 26%, Perc. 0%; — Residential density/urbanization: Obj. 10%, Perc. 26%; — Street connectivity: Obj. 7%, Perc. 29%; — Access to/availability of services and destinations: Obj. 36%, Perc. 45%; — Pedestrian and cycling infrastructure: Obj. 14% Perc. 45%; — Aesthetics and cleanliness/order: Obj. 7%, Perc. 38%; — Safety and traffic: Obj. 12%, Perc. 45%.	Active travel (AT*) categorized into: — total walking for transport — within-neighborhood walking for transport, — cycling for transport, — all AT outcomes combined (walking and cycling for transport)	Self-reported: 100%	Meta-analytic approach + quality assessment	Positive associations with all active travel for: — Walkability (*p* ≤ 0.001) — Residential density/urbanization (*p* = 0.002) — Street connectivity (*p* = 0.002) — Overall access to destinations/services (*p* ≤ 0.001) — Land use mix—Destination diversity (*p* ≤ 0.001) — Access to shops/commercial (*p* ≤ 0.001) — Access to food outlets (*p* = 0.027) — Access to business/institutional/industrial (*p* = 0.018) — Access to public transport (*p* ≤ 0.001) — Access to parks/open space/recreation (*p* ≤ 0.001) — Pedestrian-friendly features (*p* ≤ 0.001) — Availability of benches/sitting facilities (*p* = 0.004) — Streetlights (*p* = 0.013) — Easy access to building entrance (*p* ≤ 0.001) — Human or motorized traffic volume (*p* = 0.004) Negative association with total walking for transport: — Littering/vandalism/decay (*p* = 0.050)	10
Cunningham and Michael (2004)	Literature review Older adults	6 out of 27 NA	Safety Aesthetics Convenience or access to facilities Microscale urban design Land-use mix	— Self-report: 83% — Self-report + secondary data (observational): 17%	Physical activity, walking	NA	NA	Associations with physical activity for: — Low safety (unattended dogs, inadequate lighting) — Safety of footpath, lack of hills — —Noise — Aesthetics (Lack of enjoyable scenery) — Convenience to facilities/Land-use mix Association with walking: — Convenience to facilities — Safety	5
* Gadais et al. (2018)	Systematic review Elderly involved in 9% of the studies	264 (19 about seniors) Quantitative, qualitative and mixed studies, situation and literature reviews	— Land use patterns: Mixed, Density; — Urban design characteristics: Street, Site; — Transportation systems: Road network, Nonmotorized transport infrastructures, Public transport infrastructures.	*—Survey: 45% — Document: 37%, — Interview: 12%, — Focus group: 5%, — Observation 2%	Physical activity and active travel	NA	NA	* Recommendations for improving physical activity: — Facilitating and encouraging access to active transportation by means of safe and attractive infrastructure; — Easy access to multiple infrastructure/facilities through walking and cycling paths, wheelchair access, walking access, lighting, pedestrian crossings, parks and services.	8
Levasseur et al. (2015)	Scoping review Older adults	39 out of 50 Cross-sectional, longitudinal and qualitative studies	Environment categories according to International Classification of Functioning, Disability and Health (ICF): — Products and technology — Natural environment and human-made changes — Support and relationships — Attitudes — Services, systems and policies	Neighborhood measures: Objective: 14%, Subjective: 68%, Both: 18%.	Mobility	Objectively: 18%, Self-reported: 82%.	Sum of associations of single articles	Main positive association with mobility for: — Space for socialization — Seating — Aesthetics — Good condition of streets/path — Good user-friendliness of the walking environment — Proximity to resources and recreational facilities — Sidewalks — Walking/cycling facilities — Nature and green space — Street lighting — Public transportation — Neighborhood security Main negative associations with mobility for: — Poor user-friendliness of the walking environment — Traffic — Neighborhood insecurity	9
Moran et al. (2014)	Systematic reviewOlder adults ≥65 years old	31 Qualitative and mix studies	Categorization done by authors: — Pedestrian infrastructures: sidewalk characteristics, separation between pedestrians and other nonmotorized transport; — Safety: crime-related safety, traffic-related safety; — Access to facilities: access to exercise opportunities, access to daily destinations, access to rest areas; — Aesthetics: buildings and streetscape, natural scenery; — Environmental conditions: weather, environmental quality.	Indoor interviews (individual or focus groups): 68% Spatial qualitative methods: 32% — Photo-voice: 30%; — On-site observations: 30%; — Walk-along interviews: 30%; — Virtual reality route: 10%.	Physical activity	NA	Sum of associations of single articles	Possible association with physical activity for: Pedestrian infrastructure: — Sidewalk characteristics (sidewalks’ presence and continuity, quality and maintenance, slopes and curbs, and temporary obstacles on sidewalks) — Separation between pedestrians and other nonmotorized transport (cyclists, skateboarders and rollerbladers on sidewalks) Safety — Crime-related safety (lack of street lighting and upkeep) — Traffic-related safety (zebra-crossing characteristics) Access to facilities: — Access to exercise opportunities (recreational facilities, senior oriented group activities and green open space) — Access to daily destinations and public transit — Access to rest areas (benches, public washrooms) Aesthetics — Buildings and streetscape (private property, public realm) — Natural scenery (presence of greenery and water) Environmental conditions: — High environmental quality — Pollution	10
Tuckett et al. (2018)	Integrative literature review Older adults	NA NA	NA	NA	Physical activity	NA	NA	Physical activity associated with neighborhood walkability which relates to: — Convenient transit locations; — Availability of nonresidential destinations (shops, public services, places for social interaction) — Traffic — Pedestrian-friendly neighborhood, footpath quality, signaled crosswalks — Poor pedestrian access to shopping centers — Safety from crime — Scenery and places to stop and rest — Well-connected streets — Mixed land use	7
Van Cauwenberg et al.(2011)	Systematic review Older adults ≥65 years old	31 Cross-sectional and longitudinal	NEWS categories: — Walkability — Access to services — Walking/cycling facilities — Safety — Aesthetics — Urbanization	— Objective: 39% — Subjective: 42% — Both: 19%	Physical activity (PA): — Recreational PA ** — Total walking and cycling — Recreational walking — Transportation walking	Objective: 6% Subjective: 94%	Sum of associations of single articles	Results were inconsistent but most of the studied environmental characteristics that were reported were not related to PA **.	8
Van Cauwenberg et al. (2018)	Systematic review and meta-analysis Older adults ≥65 years old	72 Cross-sectional and longitudinal	NEWS categories: — Walkability — Residential density/urbanization — Street connectivity — Access to/availability of services/destinations — Pedestrian/cycling infrastructure and streetscape — Aesthetics and cleanliness/order — Safety and traffic	Total environmental attributes: Objective: 42%, Perceived: 63% Specifically: — Walkability: Obj. 15%, Perc. 0%; — Residential density/urbanization: Obj. 28%., Perc. 19%; — Street connectivity: Obj. 10%, Perc. 19%; — Access to/availability of services/destinations: Obj. 28%, Perc. 36%.; — Pedestrian/cycling infrastructure and streetscape Obj. 14%, Perc. 32%; — Aesthetic and cleanliness/order: Obj. 15%, Perc.32%; — Safety and traffic: Obj. 13%, Perc. 46%.	Physical activity during leisure-time (LTPA): — leisure-time walking — leisure-time walking within the neighborhood — leisure-time cycling — leisure-time walking and cycling combined — overall LTPA	Self-reported: 100%	Meta-analytic approach + quality assessment	Positive associations with leisure-time walking for: — Walkability (*p* = 0.01) — Land-use mix—Access (*p* = 0.02) — Aesthetically pleasing scenery (*p* < 0.001) Positive associations with leisure-time walking within the neighborhood for: — Land-use mix—Access (*p* = 0.03) — Access to public transit (*p* = 0.05) Negative association with leisure-time walking within the neighborhood for: — Barriers to walking/cycling (*p* = 0.03) Positive relationships for overall leisure time physical activity for: — Access to recreational facilities (*p* = 0.01) — Access to parks/open space (*p* = 0.04)	10
Won et al. (2016)	Systematic reviewOlder adults ≥50 years old	16 out of 32 Cross-sectional and longitudinal	Four domains of neighborhood safety: — Overall/general neighborhood safety; — Crime-related safety; — Traffic-related safety; — Proxies for safety	Total environmental attributes: Objective: 6%, Subjective: 69%, Both: 25% Specifically: — Overall/general neighborhood safety: Subj. 69%, Both 6%; — Traffic-related safety: Obj. 19%, Subj. 25%, Both 6%; — Crime-related safety: Obj. 13%, Subj. 38%; — Proxies for safety: Obj. 6%, Subj. 38%, Both 6%.	Physical activity and walking	— Physical activity: Objective 86%, Subjective 14%; — Walking: Subjective: 100%.	NA	Associations of traffic-related safety consistently significant for physical activity =Associations of crime-related safety consistently significant for walking	9
Yen et al. (2009)	Systematic reviewOlder adults ≥55 years old	7 out of 33 Cross-sectional and longitudinal	— Physical environment (commercial services, traffic, trash, neighborhood design: housing density, land-use diversity) — Perceived resources or problems (traffic, litter/trash, safety/crime, access to/quality of commercial/public services)	— Physical environment (Direct observations + administrative data): 71% — Perceived resources and/or problems (from survey data): 71%	Physical activity, walking	NA	NA	Associations with PA ** for: — Access to physical activity resources — Access to parks — Lack of footpaths perceived safe for walking Associations with walking for: — Neighborhood walkability — Density of physical activity facilities — Greater numbers of street intersections — Green and opens spaces for recreation — Higher levels of facility accessibility — Safety — New urbanism (pedestrian-friendly)	8

* AT = Active Travel; ** PA = Physical Activity.

**Table 3 ijerph-17-06127-t003:** Summary table with the associations between the elements of the built environment and the types of physical activity.

Health Behaviors	Built Environment Factors	Reference of Associations (+)	Reference of Associations (−)
Overall physical activity	Walkability	Barnett et al. (2017)Tuckett et al. (2018)	
Land-use mix	Cunningham and Michael (2004)Tuckett et al. (2018)	
Street connectivity	Tuckett et al. (2018)	
Overall access to facilities	Barnett et al. (2017)Cunningham and Michael (2004)Levasseur et al. (2015)Moran et al. (2014)Tuckett et al. (2018)Yen et al. (2009)	
Access to shops/commercial	Barnett et al. (2017)Tuckett et al. (2018)	
Poor pedestrian access to shopping centers		Tuckett et al. (2018)
Access to public transport	Barnett et al. (2017)Levasseur et al. (2015)Moran et al. (2014)Tuckett et al. (2018)	
Access to nature/parks/open space	Barnett et al. (2017)Levasseur et al. (2015)Moran et al. (2014)Yen et al. (2009)	
Access to recreational facilities	Barnett et al. (2017)Levasseur et al. (2015)Moran et al. (2014)Tuckett et al. (2018)	
Access to places for social interaction	Levasseur et al. (2015)Tuckett et al. (2018)	
Access to exercise opportunities (senior oriented group activities)	Moran et al. (2014)	
Access to rest areas—Seating Benches, public washrooms	Tuckett et al. (2018)—Levasseur et al. (2015)Moran et al. (2014)	
Pedestrian-friendly infrastructure —Footpath quality, lack of hillsFootpath quality■ Sidewalk characteristics: presence and continuity, quality and maintenance, slopes and curbs, temporary obstacles on sidewalks■ Separation between pedestrians and other nonmotorized transport	Barnett et al. (2017)Gadais et al. (2018)—Cunningham and Michael (2004)Levasseur et al. (2015)Tuckett et al. (2018)■ Moran et al. (2014)	
Poor pedestrian-friendly infrastructure—Footpath quality		Levasseur et al. (2015)—Yen et al. (2009)
Pedestrian/cycling facilities	Levasseur et al. (2015)	
Aesthetics—Greenery/Aesthetically pleasing sceneryBuildings and streetscape/Natural scenery	Levasseur et al. (2015)—Barnett et al. (2017)—Tuckett et al. (2018)Moran et al. (2014)	
Lack of aesthetically pleasing scenery		Cunningham and Michael (2004)
Crime-related safety—Street lighting	Barnett et al. (2017)Tuckett et al. (2018)—Levasseur et al. (2015)	
Crime-related safety—Unattended dogs, inadequate lightingLack of street lighting and upkeep		Levasseur et al. (2015)—Cunningham and Michael (2004)Moran et al. (2014)
Traffic-related safety —Zebra-crossing characteristicsSignaled crosswalks	Won et al. (2016)—Moran et al. (2014)Tuckett et al. (2018)	
Traffic		Levasseur et al. (2015)
High environmental quality	Moran et al. (2014)	
Pollution		Moran et al. (2014)
Noise		Cunningham and Michael (2004)
Leisure time walking,Leisure time physical activity	Walkability	Van Cauwenberg et al. (2018)	
Land-use mix—access	Van Cauwenberg et al. (2018)	
Aesthetically pleasing scenery	Van Cauwenberg et al. (2018)	
Access to public transit	Van Cauwenberg et al. (2018)	
Access to recreational facilities	Van Cauwenberg et al. (2018)	
Access to park/open space	Van Cauwenberg et al. (2018)	
Barriers to walking/cycling		Van Cauwenberg et al. (2018)
Active travel	Walkability	Cerin et al. (2017)	
Residential density/urbanization	Cerin et al. (2017)	
Street connectivity	Cerin et al. (2017)	
Overall access to facilities, destinations and services	Cerin et al. (2017)Gadais et al. (2018)	
Land-use mix—destination diversity	Cerin et al. (2017)	
Access to shops/commercial	Cerin et al. (2017)	
Access to food outlets	Cerin et al. (2017)	
Access to business/institutional/industrial destinations	Cerin et al. (2017)	
Access to public transport	Cerin et al. (2017)	
Access to parks/open space/recreation	Cerin et al. (2017)Gadais et al. (2018)	
Pedestrian-friendly infrastructure—Footpath quality, pedestrian crossing	Cerin et al. (2017)—Gadais et al. (2018)	
Pedestrian/cycling facilities	Gadais et al. (2018)	
Availability of benches/sitting facilities	Cerin et al. (2017)	
Street lighting	Cerin et al. (2017)Gadais et al. (2018)	
Easy access to building entranceWheelchair access, walking access	Cerin et al. (2017)Gadais et al. (2018)	
Human and motorized traffic volume	Cerin et al. (2017)	
Littering/vandalism/decay		Cerin et al. (2017)
Walking	Walkability	Barnett et al. (2017)Yen et al. (2009)	
Residential density/urbanization—Density of physical activity facilities	Barnett et al. (2017)Yen et al. (2009)	
Street connectivity	Yen et al. (2009)	
Overall access to facilities, destinations and services	Barnett et al. (2017)Cunningham and Michael (2004)Yen et al. (2009)	
Access to shops/commercial	Barnett et al. (2017)	
Access to public transport	Barnett et al. (2017)	
Access to nature/parks/open space	Barnett et al. (2017)Yen et al. (2009)	
Pedestrian-friendly infrastructure	Barnett et al. (2017)Yen et al. (2009)	
Greenery/aesthetically pleasing scenery	Barnett et al. (2017)	
Street lighting	Barnett et al. (2017)	
Crime-related safety	Barnett et al. (2017)Cunningham and Michael (2004)Won et al. (2016)Yen et al. (2009)

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
