# Peer review of "Impact of the Built Environment and the Neighborhood in Promoting the Physical Activity and the Healthy Aging in Older People: An Umbrella Review"

_ijerph, 2020, doi:10.3390/ijerph17176127_

Round 1

Reviewer 1 Report

Comments to authors

It is a comprehensive review of a problem previously identified by other authors in different isolated studies.

This review, combines studies and synthesizes specific elements of the built environment, which can contribute to promoting physical activity in older people.

The review contemplates only studies of scientific quality that contribute interest to the area of knowledge.

It is desirable to make a comprehensive conclusion that indicates the possible actions or alternatives necessary to provide a solution in the short or medium term. That the authors propose, to strengthen the weaknesses identified in the review, if these suggested alternatives can identify who is responsible.

From the associated and identified elements, conclude whether it is possible to include them within the solutions that concern a specific sector.

Reviewer 2 Report

First - a very interesting collective review has been developed.

Second - a detailed analysis of the data contained in 11 reviews was performed.

Reviewer 3 Report

The manuscript is interested and well written. No major remarks to be highlighted, only a couple of questions for the authors:

As an inclusion criterion for the “intervention” aspect, were there only considered cases where an additional feature was introduced or were also considered cases where a significant improvement took place? For example, if there is park in the neighbourhood, but it has been virtually abandoned by local authorities, it might not be that appealing to anyone and, so, it is almost as if it does not exist at all. Hence, if it is restored, this would not qualify (typically) as an added feature, but still it will become available and appealing to the public again. If such cases were not considered, could the authors please explain the rationale for that?

In Sub-section 2.2 reference is made to Appendix A where the results for works until the July, 31st 2019 are shown. It is the reviewer’s understanding that the findings of this search were used for the analysis in this manuscript. Is this correct? If so, as it has been a year since then, has an additional search been performed for more recent works that potentially should also be considered in the present work?
